# Progress of Research into Novel Drugs and Potential Drug Targets against Porcine Pseudorabies Virus

**DOI:** 10.3390/v14081753

**Published:** 2022-08-11

**Authors:** Mo Zhou, Muhammad Abid, Shinuo Cao, Shanyuan Zhu

**Affiliations:** 1Jiangsu Key Laboratory for High-Tech Research and Development of Veterinary Biopharmaceuticals, Jiangsu Agri-Animal Husbandry Vocational College, Taizhou 225306, China; 2Viral Oncogenesis Group, The Pirbright Institute, Ash Road Pirbright, Woking, Surrey GU24 0NF, UK

**Keywords:** pseudorabies virus, prevention, treatment, drugs, drug targets

## Abstract

Pseudorabies virus (PRV) is the causative agent of pseudorabies (PR), infecting most mammals and some birds. It has been prevalent around the world and caused huge economic losses to the swine industry since its discovery. At present, the prevention of PRV is mainly through vaccination; there are few specific antivirals against PRV, but it is possible to treat PRV infection effectively with drugs. In recent years, some drugs have been reported to treat PR; however, the variety of anti-pseudorabies drugs is limited, and the underlying mechanism of the antiviral effect of some drugs is unclear. Therefore, it is necessary to explore new drug targets for PRV and develop economic and efficient drug resources for prevention and control of PRV. This review will focus on the research progress in drugs and drug targets against PRV in recent years, and discuss the future research prospects of anti-PRV drugs.

## 1. Introduction

Pseudorabies virus (PRV), also known as suid herpesvirus type 1, belongs to the Herpesviridae family along with varicella zoster virus (VZV) and equine herpesvirus [1,2,3,4]. Pigs are the natural host of PRV, though it also infects a variety of domestic and wild animals, such as cattle, sheep, dogs, cats, and other vertebrates. Pseudorabies (PR) is a major disease that affects pig production and has a mortality rate of nearly 100% in piglets [5,6,7]. PRV causes reproductive problems in sows and boars, such as abortion, stillbirth, mummification, and fertility issues, which result in huge economic losses for the swine industry [8,9]. It was previously thought that humans had natural resistance to PRV. However, mutated PRV has been reported to cause infections in humans. In 2018, researchers confirmed the existence of human PRV infection at the genetic level, though there has been no evidence of transmission from other animals to humans or humans to humans [10,11,12,13]. Clinical cases suggest that PRV infection is possible in humans, and a stable zoonotic strain can result from a PRV mutation.

PRV is a double-stranded DNA virus with enveloped icosahedral symmetry. It is spherical in structure and has a diameter of approximately 120–200 nm. The large PRV genome contains approximately 143,000 base pairs, consisting of unique long and unique short linear nucleotide sequences [14,15]. The viral envelope consists of at least 15 proteins, 11 of which are glycosylated (gN, gM, gL, gK, gl, gH, gB, gG, gE, gC and gD). Four glycoproteins, gB, gD, and heterodimer gH/gL are required for viral replication [16,17]. Other glycoproteins including gE, gL, membrane protein US9, and non-structural protein thymine kinase (TK) are not required for viral replication but are associated with PRV virulence [18,19,20,21,22,23,24].

Presently, there is no specific treatment for PRV infection in pigs. Vaccination can fundamentally prevent the infection and transmission of the PRV and is the most effective measure for preventing and controlling PR. There is still a lack of clinical consensus and experience regarding human PRV infection because only a few confirmed cases have been reported, and patients with mild symptoms can recover without treatment. Acyclovir is recommended for treating acute retinal necrosis syndrome and encephalitis caused by VZV infection, which is also a varicella herpesvirus along with PRV. Since there are no similar treatments for PRV, it is vitally important that new drug targets should be discovered, and new antiviral drugs are being developed for the treatment, prevention, and control of PRV. This study summarizes the research results of anti-PRV drugs that have been studied in recent years and provides a theoretical basis for the development of anti-PRV drugs.

## 2. Natural Medicines Inhibit PRV Replication

Natural medicines have been widely accepted and used throughout history for the prevention of diseases, and interest in these medicines for use in the potential development of antiviral drugs has been increasing [25,26,27,28,29]. The antiviral mechanisms of natural drugs can be divided into direct inhibition and indirect inhibition. Direct inhibition refers to blocking a link during the process of viral replication, so as to remove pathogens from the body. Indirect inhibition refers to stimulating and mobilizing the immune defense system of the body to play an antiviral role [30,31,32,33,34]. The antiviral effect of natural drug compounds is better than with a single natural drug; however, natural drug compounds have numerous effective components and complex antiviral mechanisms. Previous research has indicated that some of these natural drugs have antiviral effects against PRV, providing a theoretical basis for the development of these compounds into anti-PRV drugs (Table 1).

### 2.1. (–)-Epigallocatechin-3-Gallate Inhibits PRV Infection through Inhibiting Virus Replication and Adsorption

Green tea, and its major constituent (–)-epigallocatechin-3-gallate (EGCG), has been reported to mitigate several chronic diseases [55,56]. In solid green tea extract, EGCG is the most abundant bioactive polyphenol, which has antibacterial, antiviral, and antioxidant properties [57,58,59,60,61,62]. EGCG prevents many viruses, including HCV, herpes simplex virus (HSV) and human immunodeficiency virus type 1 (HIV-1), from entering target cells, thus preventing infection [35,63]. In PK15 B6 cells and Vero cells, Huan et al. reported that EGCG significantly inhibited PRV strain infectivity [35]. They also confirmed that EGCG inhibited the adhesion, entry, and replication of PRV strains in vitro. In addition, the survival rate of PRV-challenged mice was improved significantly by EGCG. Thus, the therapeutic effect of EGCG on PRV infection in vivo was demonstrated, indicating its potential as an antiviral drug.

### 2.2. Resveratrol Inhibits PRV Multiplication in Host Cells

When plants are stimulated, resveratrol (3,5,4-trihydroxystilbene), a nonflavonoid polyphenol organic compound, is produced. Resveratrol is produced in grape leaves and is the main bioactive ingredient in wine and grape juice [36].

Resveratrol has antioxidant, anti-inflammatory, anticancer, and cardiovascular protective properties, which have been confirmed through in vitro and animal experiments [64,65,66,67,68,69]. Resveratrol possesses antiviral activity against a number of different viruses, such as herpesviruses, retroviruses, respiratory syncytial viruses, and HIV-1 [70,71,72]. Zhao et al. evaluated the antiviral activity of resveratrol against PRV and investigated the change in relevant virus-host systems after PRV infection [36]. The mortality was reduced and the growth performance of PRV-infected piglets was increased by the treatment of resveratrol. It appears that resveratrol has antiviral properties because it inhibits viral multiplication in host cells, and it relieves PRV-induced inflammation in host cells. There is a strong correlation between IκB kinase activity and NF-κB activation; an important way to determine whether NF-κB signaling is active is by measuring the levels of IκB kinases in a cell. Zhao et al. tested the IκB kinase activity of the host cells after PRV infection. The ability of resveratrol to inhibit IκB kinase activity has been confirmed to suppress both NF-κB activation and NF-κB dependent gene expression. This suggests that resveratrol may have great potential as an antioxidant and an effective anti-PRV drug.

### 2.3. Germacrone Affects the Cell Antiviral Mechanism in Early Replication

Gingeraceae plants produce germacrone, a monocyclic sesquiterpene with antitumor, antiviral, antibacterial, and anti-inflammatory properties [73,74,75]. The action of germacrone on tumors is mediated through G2/M cell cycle arrest and apoptosis promotion [73,76,77]. In addition to its ability to inhibit influenza virus (IAV) replication, germacrone can also inhibit porcine parvovirus (PPV), feline parvovirus (FPV), and porcine enterovirus (PEV) replication [75,78,79,80]. An in vitro study conducted by He et al. showed that germacrone inhibits PRV replication dose-dependently, and a further study indicated that germacrone exerts its antiviral effect during the early stage of infection [37]. It may be useful as a therapeutic drug because germacrone inhibits PRV replication by affecting the cell antiviral mechanism during the early stages of PRV replication.

### 2.4. Quercetin Inhibits PRV Infections by Decreasing the Binding Activity of gD

The most widely distributed plant flavonoid is quercetin (3,3,4,5,7-pentahydroxyflavone), which can be found in fruits, vegetables, grains, Chinese herbal medicines, teas, and wines [81]. Quercetin is a flavonoid with antioxidant, anticancer, antibacterial, and antiviral activities [82,83,84,85,86,87]. Sun et al. confirmed that quercetin inhibits PRV replication dose-dependently and is effective against a wide spectrum of PRV strains [38]. Further study confirmed that quercetin inhibited PRV infections by affecting the binding ability of gD, which is important for viral infection. From silico molecular docking data analysis, it was discovered that quercetin interacts with PRV gD-protein, thereby blocking nectin-1 binding to gD. Quercetin also reduces PRV replication in vivo, reducing mortality and decreasing the viral loads in PRV-infected mice brains. Quercetin also exhibited anti-PRV activity in vivo; mortality was reduced and the viral loads were decreased in PRV-infected mice brains. Considering quercetin’s powerful therapeutic effects against PRV infection in vitro and in vivo, it might be useful as an anti-PRV agent.

### 2.5. Curcumin Plays a Neuroprotective Role against PRV Infection

In turmeric, curcumin is the most active component of the curcuminoids, which have a variety of pharmacological properties, such as anti-inflammatory, antioxidant, anti-proliferative, and antiviral effects [88,89]. Several reports suggest that curcumin may be helpful against dengue [90], HSV [91], and vesicular stomatitis virus [92]. Newborn piglets with porcine PRV infection suffer from severe neurological disorders and die at an alarming rate. There are few drugs available to relieve the neurological symptoms caused by herpes virus. Curcumin may prove a promising treatment for neurodegenerative diseases because it crosses the blood–brain barrier [93,94]. Yang et al. explored novel drugs to relieve the neurological symptoms caused by PRV, and examined the role of curcumin in preventing oxidative stress, apoptosis, and mitochondrial dysfunction in rat hippocampal neurons infected by PRV [39]. By preventing neuronal cell death induced by various stimulants, the BDNF/TrkB pathway contributes to neuroprotection [95]. By upregulating the BDNF/TrkB pathway, curcumin improved the viability of PRV-infected hippocampal neurons and provided neuroprotection against infection by PRV. Therefore, curcumin has the potential for development as a novel drug for treating PRV infection in neurons.

### 2.6. Flos Lonicerae Japonicae Water Extract and Luteolin Suppress PRV-Induced NF-κB Pathway Activation in RAW264.7 Cells

As an antibacterial, antiviral, anti-inflammatory, antioxidative, and hepatoprotective plant, Flos Lonicerae Japonicae (FLJ) has been used in Chinese herbal medicine for over 2000 years [40,96,97,98,99,100]. When phenolic acids and flavonoids are in high concentrations, cell viability increases dramatically. Hence, flavonoids and phenolic acids present in the herbs are responsible for their antioxidative activity. Lin et al. assessed the polyphenolic profile of FLJ water extract and the antiviral effects of FLJ water extract on PRV-infected RAW264.7 T cells [40]. FLJ water extract showed antiviral activity by reducing the expression of inflammatory mediators including COX-2 and iNOS, and it reduces inflammatory cytokines by inhibiting the JAK/STAT1/3-dependent NF-κB pathway and increasing HO-1 expression. Thus, FLJ water extract can be used as a potential antiviral agent for the treatment of PRV infection.

There are high levels of luteolin (3’,4’,5,7-hydroxyl-flavone) present in many common medicinal herbs, including artichokes, celery, chamomile, and green pepper [41]. Liu et al. tested the anti-inflammatory effects of luteolin in PRV-infected RAW264.7 cells [101]. It was found that luteolin significantly inhibited NO and iNOS production, as well as COX-2 and inflammation-related cytokines. Moreover, luteolin inhibited PRV-induced NF-κB through STAT1 and STAT3 phosphorylation, but not ERK1/2, p38, or JNK1/2 induction. Luteolin could be considered a potential agent for use in treating inflammation during viral infection.

### 2.7. Platycodon Grandiflorus Polysaccharides Inhibit PRV Replication

Ancient Chinese medicinal plants like Platycodon grandiflorus have been found to have immunomodulatory, anti-inflammatory, and antiviral properties [102,103]. The polysaccharides are believed to have antiviral properties, and the antiviral properties of polysaccharides have recently gained increased attention [42,104,105,106,107]. Xing et al. investigated the antiviral activity of total Platycodon grandiflorus polysaccharides (PGPSs) against PRV and the molecular mechanism of the anti-PRV effect [42]. PGPS decreased the viral proteins in PK-15 cells 12 h post-infection significantly. They found that PGPSt affects PRV-induced autophagy, which likely explains the antiviral mechanism of PGPSt. PRV-induced autophagy was downregulated by PGPS via the Akt/mTOR signaling pathway. PRV-induced, Thus, PGPSt may inhibit PRV replication by targeting autophagy, and has the potential to be a novel antiviral drug.

### 2.8. Kaempferol can Inhibit PRV Latency

In a variety of plants, kaempferol can be found as a natural flavanol derived from ginger rhizomes [108,109,110]. As a treatment for diabetes and osteoporosis, kaempferol has attracted widespread attention due to its anticancer, anti-inflammatory, antioxidant, and antiviral properties [111,112,113,114]. Li et al. evaluated the ability of kaempferol to inhibit PRV replication in vitro and in vivo [43]. PRV replication was inhibited by kaempferol dose-dependently in PRV-infected cells and the survival rate of PRV-infected mice was increased.

They found that kaempferol inhibited PRV replication dose-dependently in vitro and improved the survival rate of PRV-infected mice in vivo. The expression of the IE180 gene, the major trans-activator of the PRV, required for the transcription of early genes. IE180, EPO, and TK transcriptions were inhibited simultaneously in the brain with the immediate-early IE180 gene. Kaempferol appears to inhibit latency-associated transcript expression in the brain. Latent infection is a difficult problem when treating herpesvirus infection. Inhibition of latent infection and virus activation are both especially important for the treatment of this disease. Based on the above data, kaempferol will be a novel alternative measure for control of PRV infection.

## 3. Small Molecules Inhibit PRV Replication

Currently, the main drug used for the treatment of HSV is acyclovir [115]. Acyclovir is a guanine nucleoside analogue, anti-DNA viral drug. Acyclovir triphosphate is phosphorylated in infected cells by HSV thymidine kinase and host cell kinase to produce acyclovir triphosphate, which competes with viral DNA polymerase inhibition or is incorporated into viral DNA, which blocks DNA synthesis [116]. However, acyclovir is associated with several drawbacks, including its short half-life and carcinogenic and embryotoxic effects. In addition, PRV strains with mutations in the viral thymidine gene have appeared, which may confer resistance to acyclovir [24]. Furthermore, acyclovir may not be able to prevent the virus from reactivating once latency was established. Therefore, research into new small molecule chemicals that can treat PRV infection is necessary (Table 1).

### 3.1. Polymers Inhibit the Attachment of PRV

A wide variety of naturally occurring and synthetic polymers exhibit antiviral properties by preventing the entry of herpesviruses into cells. Many negatively charged polymers also exhibit antiviral properties [117,118,119,120]. High antiviral activity of the synthetic polysaccharide dextran sulfate, sulfated chitin, and octa-decylribo-oligosaccharides have been characterized. A conjugate 2,5-dihydroxybenzoic acid-gelatin (2,5-DHBA-gelatin) was synthesized by Lisov et al., and the antiviral activity of this conjugate against PRV and bovine herpesvirus type 1 (BoHV-1) was evaluated [44]. The virucidal effect of 2,5-DHBA-gelatin conjugate was indirect, though it did inhibit the attachment of viruses to target cells during virus adsorption. Thus, this conjugate will be a promising synthetic polymer for antiviral formulations development against alpha herpesvirus infections since the adsorption of PRV to cells was strongly inhibited.

### 3.2. Adefovir Dipivoxil Potently Protects Mice against Lethal PRV Infection

There are 1818 kinds of drugs in the FDA-approved small-molecule library; these drugs are available for a variety of human diseases, and they are generally safe and effective. In order to develop more effective anti-PRV drugs, Wang et al. established an accurate and reliable method for screening drugs in the FDA-approved small-molecule library to determine whether they are active against PRV in vivo [45]. Several drugs inhibited PRV proliferation at nanomolar concentrations, including puromycin dihydrochloride, mitoxantrone, mitoxantrone hydrochloride, and adefovir dipivoxil. A lethal mouse model was established to confirm the effectiveness of candidate PRV drugs, in which PRV injections were administered intraperitoneal in mice. There was a noticeable reduction in clinical score in the adefovir dipivoxil treated mice compared to the untreated mice after they were treated with adefovir dipivoxil. Therefore, adefovir dipivoxil might offer important guidance for identifying candidate drugs with the potential to control PRV epidemics.

### 3.3. Valproic Acid Derivative Inhibits PRV Infection 

Currently there are no antiviral drugs for treating PRV infection, thus, new drugs are urgently needed. Acyclovir is a guanosine analogue, which interferes with viral DNA replication. However, as previously mentioned, acyclovir does have some advantages [121,122,123]. Valproic acid (VPA) is the alternative of acyclovir; it can interrupt the infectious cycle of several enveloped viruses. VPA also has hepatotoxic and teratogenic activity [124,125,126]; however, derivatives of VPA have been shown to be less toxic compared to VPA. To determine whether valpromide (VPD), a compound derived from VPA, has an antiviral effect against PRV infection, Andreu et al. tested the PRV-infected PK15 swine cell line and the Neuro-2a neuroblastoma cell line [46]. The inhibitory effect of VPD on PRV infection was similar with ACV in the PRV-infected cell lines. Therefore, in the future, VPD will be a viable alternative to nucleoside analogues for treating PRV-related diseases, but the pharmacokinetic study and the antiviral mechanism of VPD still need to be invested. 

### 3.4. Hydroquinone Inhibits PRV Replication in Neurons In Vitro and In Vivo

The chemical compound hydroquinone has a variety of biological effects. High concentrations of hydroquinone are able to cause cell cycle arrest, induce apoptosis, and promote cell death in a variety of ways [47,127,128]. When macrophage-mediated inflammation occurs, hydroquinone can target AKT and promote its phosphorylation [129]. Based on a screening of 44 FDA-approved drugs, Fang and colleagues found hydroquinone to be highly anti-PRV active, inhibiting PRV adsorption on cell surfaces and internalization [47]. Further research found that the PRV inhibition induced by hydroquinone was related to AKT phosphorylation. Based on a in vivo experiment, they found that the viral loads in tissues and mortality in PRV-infected mice was reduced significantly in hydroquinone treated mice. Thus, hydroquinone will be an excellent therapeutic agent for the treatment of PR.

### 3.5. Diazadispiroalkane Derivatives Block the Attachment of PRV

Human cytomegalovirus (HCMV) is an opportunistic pathogen that can cause serious illness, including death [130,131,132]. Most anti-HCMV drugs, such as ganciclovir and cidofovir, inhibit the viral DNA polymerase enzyme. However, these inhibitors have some drawbacks, a low molecular weight inhibitor, N-N-(bis-5-nitropyrimidyl) dispirotripiperazine derivative (DSTP-27), is a new class of non-nucleosidic antiviral agent against HSV-1 and HCMV, but DSTP-27 showed metabolic instability associated with nitric oxide release in vivo [133]. Two new DSTP-27 derivatives synthesized by Adfeldt et al., and the antiviral activity of these two derivatives against HCM and PRV at each stage of infection was studied [48]. Based on these results, the new derivatives of DSTP-27 may prove to be promising candidates to prevent viral attachment to cell surfaces via binding to heparan sulfate glycosaminoglycans (HS).

### 3.6. Ivermectin Inhibits PRV Proliferation In Vitro and In Vivo

In recent years, a small molecule macrocyclic lactone known as ivermectin, which has been approved for parasitic infections by the US Food and Drug Administration, has received renewed interest in recent years, because of its apparent potential as an antiviral drug [134]. Based on the fact that ivermectin binds to the importin α (IMPα) protein and inhibits its nuclear transport role, ivermectin appears to have broad antiviral activity. Ivermectin exhibits antiviral potential inhibits on HIV-1 and DEV proliferation by restraining nuclear transport of integrase and the non-structural protein 5 polymerase [135]. Lv et al. determined that ivermectin has antiviral effects against PRV and studied the mechanism(s) involved in this antiviral activity [49]. As a result of the treatment with ivermectin, UL42 localization at the nucleus was disrupted and subsequent progeny virus production was significantly decreased. When PRV-infected mice were treated with ivermectin, the survival rate of mice was significantly increased, and the infection was relieved, the clinical scores were lower and the gross lesions in the brain were fewer compared with the untreated mice. Since UL42 of PRV requires importin-α/β-mediated nuclear import pathways for nuclear transport, ivermectin could serve as an anti-PRV drug candidate.

## 4. Application of New Technology and Materials in Anti-PRV Drug Research

The PRV has caused significant damage to the swine husbandry industry and poses a potential threat to humans. However, vaccine research has progressed relatively rapidly, resulting in the development of a vaccine against PRV. There has been much progress made in the development of vaccines against PRV, though only a few antiviral agents against PRV were available in recent years. Additionally, several novel approaches have also been used for PRV containment, including DNA replication inhibitors and viral reverse transcriptase inhibitors (Table 1).

### 4.1. 3D8 scFv Prevents PRV Infection in Mice

Single chain variable fragments of 3D8 are recombinant monoclonal antibodies with nuclease activity. They were originally isolated from autoimmune-prone mice and exhibit non-specific activity on DNA and RNA [136]. They are active against HSV, PRV, CSFV, murine norovirus (MNV), 2 geminiviruses, 5 tobamoviruses, and cucumaviruses [137,138,139]. Lee et al. purified the 3D8 scFv protein from *E. coli* and evaluated the antiviral effects in PRV-infected mice [50]. The 3D8 scFv is transmitted virtually to all organs of the mouse, localized to the cytoplasm by caveolae-mediated endocytosis, and inhibited the virus in all organs, especially in the brain. A further analysis was carried out to determine whether the 3D8 scFv damaged the viral genome directly or indirectly by inducing gene expression levels in the inflammatory pathway. The results demonstrated that 3D8 scFv induced the antiviral effects by itself directly. As the study indicates, the 3D8 scFv is a broad-spectrum, multifaceted antiviral agent that can digest viral genomes without sequence specificity, thereby inhibiting the spread of viruses, and could be used to explore novel drugs against DNA and RNA viruses in the future.

### 4.2. Targeting UL42 by RNAi Efficiently Inhibits PRV Replication

Gene silencing through RNA interference (RNAi) is a conserved mechanism involving small interfering RNAs that cause homologous RNAs to degrade sequence-specifically [140]. Antiviral therapies based on RNAi have shown to be novel and effective against many different viruses. As UL42 is a processivity factor for PRV, it enhances the catalytic activity of DNA polymerase and is essential for viral replication. Therefore, UL42 may be an antiviral target [52,141]. A group of researchers synthesized three siRNAs directed against UL42 in cell culture and investigated their antiviral properties [52]. The siRNAs induced potent inhibition of UL42 expression after PRV infection and decreased PRV replication. The RNAi technique may provide new clues for designing intervention strategies against herpesviruses by targeting their processivity factors.

### 4.3. Polyvalent 2D Entry Inhibitors Inhibit PRV Entrance

There are many biological processes in nature that involve noncovalent interactions between ligands and receptors, such as adhesion, cell-to-cell recognition, and self-organization. Among the most interesting phenomena in this context is how viruses adhere to a target cell surface, and how they enter the cell via endocytosis or cell fusion. Inhibitors targeting this process can effectively bind viral particles and block their contact with surfaces, thus preventing viral adhesion and infection. According to Haag et al., a polyglycerol sulfate-functionalized graphene sheet was synthesized and evaluated for use as an extracellular matrix-inspired entry inhibitor [51]. By binding enveloped viruses during adhesion, the developed 2D architectures demonstrated strong inhibitory activity against African swine fever virus (ASFV) and PRV. Compared to enrofloxacin and heparin, the developed 2D architectures had equivalent or better inhibitory properties against PRV. Thus, the developed polyvalent 2D entry inhibitors could serve as efficient entry inhibitors for other enveloped viruses. The concept of the 2D molecule designs represents a novel strategy for developing novel drugs against enveloped viruses.

### 4.4. Natural Polypeptide Inhibits of PRV Proliferation

Antimicrobial peptide (AMP) is an effective natural polypeptide, which has gained worldwide attention as an alternative to antibiotics [53,142,143,144,145]. AMP is a new type of therapeutic agent developed for both its antibacterial and antiviral activities [122,125]. Piscidin 1 belongs to the piscidin family. It was discovered in the mast cells of fish. AMPs isolated from other hosts have been found to exert antiviral activity and can inhibit infections with enveloped viruses, such as human cytomegalovirus (HCMV), HSV-1, HSV-2, and vesicular stomatitis virus (VSV) [146]. Therefore, AMPs may be promising agents for use against herpesvirus infections. A research was performed to study the pharmacokinetics of piscidin 1 by Hu et al., [53] and the antiviral activity of piscidin was confirmed. In an evaluation of five broad-spectrum AMPs against PRV, Hu et al. found that piscidin had the strongest antiviral activity due to direct interactions with the viral particles, therefore, the cells were protected from PRV-induced apoptosis. Investigation into the protective effect of piscidin in vivo showed that piscidin reduced the mortality rate of PRV-infected mice. This result suggests that piscidin-1 could be a future alternative antibiotic and/or antiviral for use in clinical veterinary medicine.

Defensins are cationic peptides, a type of antimicrobial peptide. Defensins possess disulfide bonds, are widely distributed in fungi, plants, and animals and are crucial regulatory molecules in the biological immune system [147]. In many cases, defensins can kill enveloped viruses, including HIV, herpes, and vesicular stomatitis viruses [147]. These defensins work primarily by binding to the viral coat protein; this special mechanism of action also makes it difficult for microorganisms to develop resistance to them. Defensins can directly inhibit the virus, and the degree of inhibition depends on the defensin concentration and the tightness of the intramolecular disulfide bond [148]. The antiviral activity of defensin is strong under neutral and low ionic strength conditions [149]. The peptide porcine β-defensin 2 (PBD-2) has been synthesized and its antiviral activity against PRV has been evaluated both in vivo and in vitro by Huang et al. [54]. In PK15 cells, PBD-2 inhibited PRV proliferation, resulting in improved survival of PRV-infected mice. Huang et al. generated C57/BL TG mice with PBD levels 250% higher than in WT mice by overexpressing the PBD-2 gene. When C57/BLTG mice were challenged with PRV, they developed fewer lesions in their brains, spleens, and livers compared with WT littermates, while their PRV viral loads in their brains, livers, and lungs were lower. As a result, PBD-2 could provide new therapeutic and prophylactic treatment for Aujeszky’s disease as well as other viral diseases.

## 5. Potential Targets of Anti-PRV Drugs

### 5.1. NF-κB and MAPK Signaling Pathways

Infections with pathogens often cause an inflammatory response in the host. To survive in the host cell, pathogens have evolved elaborate ways to destroy the host cell’s innate defenses. Pathogen-associated molecular patterns (PAMPs) are recognized by extracellular and intracellular pattern recognition receptors (PRRs), which stimulate a cascade of inflammatory signals. The mitogen-activated protein kinase (MAPK) and nuclear factor-κB (NF-κB) signaling pathways are activated through signal transduction when the hosts are infected by the pathogen, thus transcription of downstream anti-infection-related genes and inflammatory factors were initiated [150,151]. It has been shown that pathogen effector proteins can interfere with host inflammation by blocking MAPK and NF-κB signaling pathways during the interaction between the pathogen and host. Animals infected with PRV often exhibit inflammation, cytokine release, and peroxide production; these are essential defense mechanisms [3,152]. Uncontrollable inflammatory responses and excessive inflammatory cytokines may compromise the integrity and function of immune cells. As an animal’s immune system weakens, PRV invades its nervous system, causing neurological symptoms [3,4]. Based on the above information, a large number of studies have concluded that the NF-κB and MAPK signaling pathways are potential targets for drugs to treat PRV infection.

Degradation of IκB is a crucial step in activating the NF-κB signaling pathway, the relative levels of IκB kinases are used as the indicator of NF-κB signaling pathway activating. NF-κB responsive genes are expressed more strongly following PRV-induced degradation of IκB and translocation to the nucleus of RelA, which binds to promoter regions of these genes [153]. Resveratrol, a non-flavonoid polyphenol compound, can attenuate the degradation of IκBα and the translocation of RelA to the nucleus in PRV-infected PK15 cells, therefore affecting virus production and infection. In addition, in vivo experiments indicated that resveratrol decreased mortality, enhanced growth performance, inhibited viral reproduction, alleviated tissue inflammation and lesions, and improved the levels of cytokines in PRV-infected piglets [152]. The inhibitory effect of resveratrol on PRV proliferation can be attributed to its immunomodulatory effects of IFN-γ.

Inflammatory genes such as iNOS and COX-2 are activated by phosphorylated NF-κB p65 in the nucleus. According to Lin et al. and Liu et al., the FLJ water extract and luteolin suppressed the gene expression of inflammatory cytokines, COX-2 and iNOS, through the inhibition of JAK/STAT signaling [40]. Furthermore, the researchers discovered that exogenous antioxidants inhibited virus proliferative activity in vitro and reduced PRV-induced tissue damage by inhibiting oxidative stress. Additionally, using ethyl acetate fractions of polygonum hydropiper L. flavonoids, Ren et al. found a significant inhibition of phosphorylation and degradation; they also found that IκBα and NF-κB p65 is retained in cytoplasm and decreased in nuclei [154]. These results suggest that the NF-κB signaling pathway could be a novel target for the anti-PRV drugs. It is well known that JNK1/2, ERK1/2, and p38 MAPK are all critical kinases in MAPK signaling pathways that regulate macrophage survival and the production of inflammatory mediators. Previous studies have shown that PRV infection increases phosphorylation of ERK1/2 and p38, treatment with ethyl acetate inhibits phosphorylation of this important kinase. Therefore, a novel anti-inflammatory role for FEA on PRV via Inhibition of the ERK/MAPK Signaling Pathway. The main drug used for the treatment of HSV is acyclovir [115]. Acyclovir is a guanine nucleoside analogue, anti-DNA viral drug. Acyclovir triphosphate is phosphorylated in infected cells by HSV thymidine kinase and host cell kinase to produce acyclovir triphosphate, which competes with viral DNA polymerase inhibition or is incorporated into viral DNA, which blocks DNA synthesis [116]. However, acyclovir is associated with several drawbacks, including its short half-life and carcinogenic and embryotoxic effects. In addition, PRV strains with mutations in the viral thymidine gene have appeared, which may confer resistance to acyclovir [24]. Furthermore, acyclovir may not be able to prevent the virus from reactivating once latency was established. Therefore, research into new small molecule chemicals that can treat PRV infection is necessary.

### 5.2. BDNF/TrkB Signaling Pathway

In cultured cells, PRV causes apoptosis, which is associated with neurodegenerative diseases and neuropathies. PRV infection causes oxidative stress, as well as damage to neurons, which are the targets of neuroprotective agents [155]. A major function of brain-derived neurotrophic factor (BDNF) is to modulate neuronal plasticity, mitochondrial transport, and neuronal protection, such as anti-apoptosis, anti-oxidation, and suppression of neurodegeneration. The function of BDNF is primarily regulated by the binding of transmembrane tropomyosin-related kinase B (TrkB) to the mitochondrial membrane [95]. BDNF/TrkB contributes to the neuroprotection of neurons by preventing the death of neurons induced by various stimuli [155].

A neuroprotective agent, BDNF binds to TrkB receptors to rescue neurons from various insults. BDNF plays an important role in mitochondrial dysfunction pathophysiology and treatment. Infection with PRV may also affect levels of BDNF in hippocampal neurons [95]. Therefore, the BDNF/TrkB signaling pathway may be a drug target for the treatment of PRV infection. A study by Yang et al. found that curcumin mediated neuroprotection against PRV infection through BDNF/TrkB signaling [39]. This research suggests that the BDNF/TrkB pathway is a potential drug target for the treatment of neurological disorders caused by PRV infection.

### 5.3. Akt/mTOR Signaling Pathway

One of the most conserved physiologic functions of cells is autophagy, a process in which damaged organelles and pathogenic microorganisms are degraded [77,156,157]. Antiviral drugs may be able to target autophagy; the degradative process of cells is essential for adjusting to dynamic environments and coping with developmental changes. Autophagy is involved in cell regulation during the immune response. In many studies, it has been shown that low glucose concentrations inhibit the Akt/mTOR pathway involved in virus replication [42,94]. PRV infection has been found to significantly reduce the phosphorylation levels of Akt and mTOR and increase autophagosome formation in PK15 cells. PGPSs inhibited the PRV infection-induced autophagy and ameliorated PRV-suppression of the Akt/mTOR signaling pathway [42].

### 5.4. The Critical Factors of Attachment, Entry, and Replication

The life cycles of human and animal alpha herpesviruses are based on several major factors. When an alpha herpesvirus infects a cell, it attaches to the cell surface, fuses with plasma membrane, and enters the nucleocapsid [158,159,160]. The viral proteins and/or the interacting host proteins during the viral life cycle can be targets for drugs. Heparan sulfate proteoglycan on the cell surface is used by PRV for initial attachment via non-essential gC, which is critical for PRV attachment [161]. According to Adfeldt et al., two new diazadispiro-alkane derivatives are antiviral agents targeting gC’s primary attachment to heparan sulfate [138]. The PRV virion envelope glycoprotein D (gD) is an essential co-factor for virus entry and is responsible for recognizing and binding the virus to specific cellular receptors. As gD binds its receptors, conformational changes occur in gD, which in turn activate a multi-glycoprotein complex for triggering viral replication [162,163]. According to the virucidal properties of quercetin and the fact that it inhibits PRV infection adsorption, quercetin may play a role in the interaction between these molecules and viral particles. Silico study indicates that quercetin might interact with the gD-protein on the surface of PRV, thereby inhibiting its replication. Furthermore, according to Lisov et al., 2,5-dihydroxybenzoic acid-gelatin conjugate significantly inhibits PRV adsorption to cells and impairs PRV attachment to target cells [44].

A number of antiviral drugs are being developed to target RNA-dependent RNA polymerase and viral DNA polymerase. PRV DNA polymerase has two subunits, UL30 and UL42. A catalytic subunit with inherent polymerase activity is UL30, while a processivity factor, UL42, enhances DNA-binding specificity and decreases dissociation between DNA polymerase and viral DNA. RNAi reduces PRV replication by reducing UL42 expression, indicating that this protein is important for virus replication in the nucleus. In mice and BHK cells, Lv et al. found that sub-cytotoxic doses of ivermectin suppressed PRV replication [140]. In mice, ivermectin prevented the translocation of UL42, an accessory subunit of DNA polymerase, resulting in a reduction in lesion severity caused by PRV infection. Thus, UL42 is a potential target for antiviral drug therapy against PRV infection.

## 6. Conclusions

There are limited kinds of drugs to treat herpesvirus infection. The development of anti-herpesvirus drugs has made some progress, and a large number of natural antiviral compounds have been discovered during the development of chemical synthesis technology. Natural products come from a wide range of sources, which are easy to obtain and have low toxicity. Therefore, natural products are important resources to search for antiviral active ingredients. In addition, screening new small molecule drugs and exploring genetically engineered drugs are the main direction of anti-PRV drug development. However, the research on novel anti-PRV drugs is still in the initial stage and lacks systematic studies. Most of the antiviral activities of candidate drugs have been carried out in cell cultures, and only a few drugs have been tested in mice, and almost none in pigs. Currently, anti-PRV drug targets mainly focus on host inflammatory response related pathways, autophagy related pathways and viral replication enzymes. These studies will provide a theoretical basis for the green prevention and control of PRV and other herpesviruses.

Based on the current status of PRV drug research, the development of natural antiviral active ingredients and the screening of new small molecule drugs will be the main direction of anti-PRV drug research. The development of effective drug targets and the study of antiviral mechanisms are also the key to future anti-PRV drug research.

## Figures and Tables

**Table 1 viruses-14-01753-t001:** The reported anti-PRV drug.

Drug Types	Drug Names	Potential Targets	References
Natural medicines	Epigallocatechin-3-gallate	Undefined	[35]
	Resveratrol	NF-κB signaling pathways	[36]
	Germacrone	Undefined	[37]
	Quercetin	gD-protein	[38]
	Curcumin	BDNF/TrkB signaling pathway	[39]
	Flos Lonicerae Japonicae water extract	NF-κB signaling pathway	[40]
	Luteolin	NF-κB signaling pathway	[41]
	Platycodon grandiflorus polysaccharides	Akt/mTOR signaling pathway	[42]
	Kaempferol	Undefined	[43]
Small molecules	2,5-dihydroxybenzoic acid-gelatin	Undefined	[44]
	Adefovir dipivoxil	Undefined	[45]
	Valproic acid derivative	Undefined	[46]
	Hydroquinone	Undefined	[47]
	Diazadispiroalkane derivatives	Heparan sulfate glycosaminoglycans	[48]
	Ivermectin	UL42	[49]
Novel materials	3D8 scFv	Undefined	[50]
	Polyvalent 2D	Undefined	[51]
	RNAs	UL42	[52]
	Natural polypeptide	Undefined	[53,54]

## Data Availability

The data presented in this study are available in the insert article.

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
