# Peer review of "Progress of Research into Novel Drugs and Potential Drug Targets against Porcine Pseudorabies Virus"

_viruses, 2022, doi:10.3390/v14081753_

Round 1
Reviewer 1 Report
This review focuses on summarizing the research progress of drugs and drug targets against PRV in recent years. In general, this is an interesting review that highlights the advancement of drugs against PRV infection and discusses the future research prospects of anti-PRV drugs. However, there are several issues that need to be addressed. These points are summarized below.
1. Line 12-13, “At present, the prevention of PRV is mainly through vaccination, there are few specific antivirals against PRV, but it is not possible to treat PRV infection effectively with drugs”. In this sentence, the “not possible” should be written “possible”.
2. Line 22, “porcine herpesvirus type 1” should be “suid herpesvirus type 1”.
3. Line 57, “Natural” should be “natural”.
4. Line 79, “in vitro” should be written in italics; Line 81, “in vivo” should be in italics.
5. Line 216-217, antimicrobial peptide (AMP) is an effective natural polypeptide, which belongs to large molecules. Hence, the section of 3.2 should not be classified into “Small molecules inhibiting PRV infection”.
6. Line 348, porcine defensin is cationic peptide, which is also a type of antimicrobial peptide. So the section 4.4 and 3.2 should be combined into one part.
7. It is recommended to draw a table to present these anti-PRV drugs.
Author Response
Authors’ response to Reviewer Comments
Reviewer 1:
This review focuses on summarizing the research progress of drugs and drug targets against PRV in recent years. In general, this is an interesting review that highlights the advancement of drugs against PRV infection and discusses the future research prospects of anti-PRV drugs. However, there are several issues that need to be addressed. These points are summarized below.
Author response: We greatly appreciate Reviewer #1 for the positive comments and helpful suggestions. We have revised the manuscript as you suggested.
Specific comments
1.Line 12-13, “At present, the prevention of PRV is mainly through vaccination, there are few specific antivirals against PRV, but it is not possible to treat PRV infection effectively with drugs”. In this sentence, the “not possible” should be written “possible”.
Author response: The statement has been revised as suggested.
2.Line 22, “porcine herpesvirus type 1” should be “suid herpesvirus type 1”.
Author response: The statement has been revised as suggested.
3.Line 57, “Natural” should be “natural”.
Author response: The word has been revised as suggested.
4.Line 79, “in vitro” should be written in italics; Line 81, “in vivo” should be in italics.
Author response: The word has been revised as suggested.
5.Line 216-217, antimicrobial peptide (AMP) is an effective natural polypeptide, which belongs to large molecules. Hence, the section of 3.2 should not be classified into “Small molecules inhibiting PRV infection”.
Author response: Thanks for your critical comment, the section of 3.2 has been revised as suggested.
6.Line 348, porcine defensin is cationic peptide, which is also a type of antimicrobial peptide. So the section 4.4 and 3.2 should be combined into one part.
Author response: Thanks for your helpful suggestions,the section 4.4 and 3.2 should be combined into one part in the revised manuscript.
7.It is recommended to draw a table to present these anti-PRV drugs.
Author response: Thanks for your helpful suggestions, an additional table present these anti-PRV drugs have been drawn as suggested.

Reviewer 2 Report
The paper revealed that progress of research into novel drugs and potential drug targets against porcine PRV. However, I have several comments which need be addressed.
1. Line 13: it is not possible should be it is possible.
2. lines 14-15: but their efficacy is not significant and basic research materials are lacking.
This statement does not accord with the reported facts.
3. Some references are wrong. Such as: line 78: reference 45.... Please check the references carefully.
Author Response
Authors’ response to Reviewer Comments
Reviewer 2:
The paper revealed that progress of research into novel drugs and potential drug targets against porcine PRV. However, I have several comments which need be addressed.
Author response: The authors greatly appreciate Reviewer #2 for the critical comments and helpful suggestions. And the author has revised the manuscript as you suggested.
Specific comments
- Line 13: it is not possible should be it is possible.
Author response: Sorry for the mistake, the statement has been revised as suggested.
- lines 14-15: but their efficacy is not significant and basic research materials are lacking. This statement does not accord with the reported facts.
Author response: Thanks for your critical comment, the statement has been revised as suggested.
- Some references are wrong. Such as: line 78: reference 45.... Please check the references carefully.
Author response: Sorry for the mistake, and the wrong reference 45 has been changed.
